# AI-Based Cancer Detection Model for Contrast-Enhanced Mammography

**DOI:** 10.3390/bioengineering10080974

**Published:** 2023-08-17

**Authors:** Clément Jailin, Sara Mohamed, Razvan Iordache, Pablo Milioni De Carvalho, Salwa Yehia Ahmed, Engy Abdullah Abdel Sattar, Amr Farouk Ibrahim Moustafa, Mohammed Mohammed Gomaa, Rashaa Mohammed Kamal, Laurence Vancamberg

**Affiliations:** 1GE HealthCare, 283 Rue de la Miniére, 78530 Buc, France; 2Baheya Foundation for Early Detection and Treatment of Breast Cancer, El Haram, Giza 78530, Egypt; 3National Cancer Institute, Cairo University, 1 Kasr Elainy Street Fom Elkalig, Cairo 11511, Egypt; 4Radiology Department, Kasr El Ainy Hospital, Cairo University, Cairo 11511, Egypt

**Keywords:** contrast-enhanced mammography, breast cancer, cancer detection, deep learning, computer aided detection

## Abstract

Background: The recent development of deep neural network models for the analysis of breast images has been a breakthrough in computer-aided diagnostics (CAD). Contrast-enhanced mammography (CEM) is a recent mammography modality providing anatomical and functional imaging of the breast. Despite the clinical benefits it could bring, only a few research studies have been conducted around deep-learning (DL) based CAD for CEM, especially because the access to large databases is still limited. This study presents the development and evaluation of a CEM-CAD for enhancing lesion detection and breast classification. Materials & Methods: A deep learning enhanced cancer detection model based on a YOLO architecture has been optimized and trained on a large CEM dataset of 1673 patients (7443 images) with biopsy-proven lesions from various hospitals and acquisition systems. The evaluation was conducted using metrics derived from the free receiver operating characteristic (FROC) for the lesion detection and the receiver operating characteristic (ROC) to evaluate the overall breast classification performance. The performances were evaluated for different types of image input and for each patient background parenchymal enhancement (BPE) level. Results: The optimized model achieved an area under the curve (AUROC) of 0.964 for breast classification. Using both low-energy and recombined image as inputs for the DL model shows greater performance than using only the recombined image. For the lesion detection, the model was able to detect 90% of all cancers with a false positive (non-cancer) rate of 0.128 per image. This study demonstrates a high impact of BPE on classification and detection performance. Conclusion: The developed CEM CAD outperforms previously published papers and its performance is comparable to radiologist-reported classification and detection capability.

## 1. Introduction

Breast cancer is one of the leading causes of death related to cancer among women worldwide [1]. Early detection and accurate diagnosis are crucial for the successful treatment of breast cancer. Medical imaging techniques, such as mammography, ultrasound, and magnetic resonance imaging (MRI), are widely used for the detection and diagnosis of breast cancer. These imaging modalities have contributed significantly to the improvement of breast cancer outcomes [2].

Recently, deep learning (DL) has emerged as a promising technique in the field of medical imaging [3,4,5]. Such algorithms have demonstrated clinically relevant performance for breast imaging applications, including detection and diagnosis of breast pathologies [6,7]. Deep learning techniques can be trained on large datasets of breast images to recognize patterns and features associated with breast cancers, providing radiologists with automated tools to improve screening and diagnostic accuracy and efficiency. Multiple DL applications have emerged in the breast imaging environment, e.g., triage of normal cases (versus non-normal), lesion detection, and lesion segmentation.

Contrast-enhanced mammography (CEM) is a relatively new modality that has gained significant attention in the field of breast imaging [8,9,10,11,12]. CEM combines the advantages of digital mammography and the functional information of contrast-enhanced imaging to provide high-quality information about breast tissue [13]. A typical CEM exam starts with the administration of an intravenous injection of an iodinated contrast material, followed by multi-view image acquisitions of both breasts using low-energy (LE) and high-energy (HE) X-ray beams. The LE images are comparable to standard mammograms, which provide morphological information of the breast structures. In addition, LE and HE images are combined to provide a third image, named the recombined image (REC), which highlights the iodine contrast uptake and associated angiogenesis phenomenon, indicating potential areas of malignancy. Compared to mammography alone, CEM has been shown to improve sensitivity and specificity for breast cancer detection, particularly in women with dense breast tissue [14].

The development of artificial intelligence and computer-aided diagnostics (CAD) for CEM offers the potential to enhance diagnostic quality and workflow. Developing tools for lesion and patient-level classification, as well as detecting specific radiological signs, presents a significant opportunity to help clinicians.

## 2. Related Works on CEM-AI

In the field of CEM, current advancements in artificial intelligence for CAD are primarily focused on lesion classification to predict pathology results (e.g., cancer vs. no cancer, invasive vs. non-invasive cancer). However, due to the limited size of the available databases, typically containing a few hundred patients, most approaches rely on manually identifying and extracting regions of interest (ROI) for classification. These ROIs are then used to extract handcrafted features (such as radiomics [15,16,17,18]) or deep learning features (obtained from a convolutional neural network [19,20]), which are subsequently fed into machine learning algorithms (e.g., support vector machine, linear discriminant analysis, multi-layer perceptron, etc.) to perform classification. Such analyses require the suspicious area to be manually detected and extracted by the user. In certain scenarios, a manual pixel segmentation of the contrast uptake may be necessary [19,21]. Without manual extraction of the region of interest, DL-based breast classification has been recently studied in the literature. Li et al. developed a DL architecture with attention mechanism to classify a breast malignancy using all available views of the patient exam [22]. This study was developed from 122 patients.

Already developed in the field of contrast-enhanced MRI [23,24], its application to CEM has only recently been explored in the literature. Existing studies in this area have used various state-of-the-art DL architectures for detection and segmentation tasks, but they have been limited by small datasets. In their paper [25], Khaled et al. used a Mask-RCNN [26] architecture to detect and segment breast lesions based on a dataset of 326 patients. In a previous study [27], a detection algorithm was applied for the detection of lesions (benign and malignant) from a dataset of 536 patients. A major limitation of the two previous studies is the small number of patients in the dataset. In a recent publication from Zheng et al. [28], 1912 Asian patients where enrolled in a DL segmentation and classification study based on a Faster-RCNN [29] architecture. With promising results, this work excluded the presence of multiple, bilateral, and non-mass lesions and lesions seen only in one view that may represent complex cases. The evaluation of the impact of the image inputs and performance regarding breast clinical factors such as background parenchymal enhancement was not explored. Based on the segmentation of lesions, a complementary analysis of this study is to evaluate lesion detection metrics (e.g., number of false positive).

In this paper, we present a novel deep-learning contrast-enhanced mammography (CEM) CAD algorithm designed for the detection and classification of breast lesions. This work introduces several contributions. First, a large CEM dataset consisting of 1673 cases and 7443 pairs of low-energy (LE) and recombined (REC) images containing diverse lesion characteristics was used for training, validation, and testing the CAD model.

Secondly, the CAD model’s performance was evaluated using different clinically relevant image inputs, comparing the use of recombined image alone versus a combination of Low Energy and recombined images. This assessment allowed the optimization of the model’s effectiveness for detecting and classifying breast lesions.

The optimized CAD model’s performance was compared to state-of-the-art clinical diagnostic performance conducted by radiologists. This comparison highlighted the effectiveness of the CAD algorithm as an advanced tool for breast lesion diagnosis.

Finally, a comprehensive evaluation of the CAD algorithm was conducted for different background parenchymal enhancement (BPE) sub-classes, providing a more detailed analysis of the model’s performance under various BPE conditions. The utilization of a large dataset, diverse lesion sizes and types, and the evaluation under various BPE conditions enhance the robustness and applicability of the CAD model for improved breast cancer diagnosis.

## 3. Materials

This retrospective study leverages two distinct CEM datasets containing 1673 patients, 7443 image pairs (low-energy and recombined).
The first dataset was obtained from Baheya Foundation For Early Detection and Treatment of Breast Cancer and included CEM DICOM images, radiology, pathology, and follow-up reports, as well as clinical information, including breast density, BPE, patient age, lesion type, and lesion location annotations (boxes) provided by radiologists.The second dataset was collected from various institutions and consisted of CEM DICOM images, pathology outcome, and lesion location annotations (boxes) provided by radiologists or GE HealthCare (GEHC) CEM experts.

### 3.1. Clinical CEM Database from Baheya

1087 patients were enrolled in this retrospective study from December 2019 to December 2022. All data were acquired at Baheya Foundation For Early Detection And Treatment of Breast Cancer, Giza, Egypt. The study protocol was approved by the Institutional Review Board and informed written consent was obtained for the use of the enrolled individuals. Examinations were performed with a Senographe Pristina™ (GE Healthcare, Chicago, IL, USA).

Each patient case consisted of RAW and processed CEM images, including LE images, HE images, and REC images, on at least four standard views (right/left, cranio-caudal (CC), and mediolateral oblique (MLO)). Radiology reports containing CEM BIRADS and pathology reports were included in cases where biopsy was performed. The pathology outcome of every lesion in the dataset have been confirmed by biopsy. Non-biopsied cases, rated BIRADS 1–2, are assumed to be benign/normal. The number of normal/benign, benign biopsied, and malignant cases are, respectively, 136, 199, and 752 at the patient level.

A group of experienced radiologists from Baheya, who are specialized in breast imaging and CEM, conducted lesion location annotation using the patients’ complete medical records. The annotation process consists in marking rectangular boxes or ellipses on the biopsied areas of each view. To standardize the annotations, all ellipses were treated as circumscribed rectangular boxes.

### 3.2. Second CEM Datasets

The second CEM dataset contains 586 patients with 2510 recombined images collected from different hospitals and acquired with various imaging systems (Senographe DS™, Essential™ and Pristina™ from GE Healthcare, Chicago, IL, USA).

Each patient case consisted of RAW and processed CEM images of left/right views, mainly with both CC and MLO and associated pathology outcome. Other relevant clinical data were punctually available, such as pathology reports. The pathology outcome of every lesion in the dataset was confirmed by biopsy. Non-biopsied cases, rated BIRADS 1–2, are assumed to be benign/normal. The number of non-biopsied, biopsied benign, and malignant cases are, respectively, 191, 149, and 246 at the patient level.

Radiologists conducted enhanced lesion location annotation using all the clinical data available for each case.

### 3.3. Dataset Summary and Split

A summary of all collected cases can be found in Table 1.

To train the DL CEM CAD algorithm, the dataset was split into three parts, namely train, validation, and test sets. The test set consists of 150 cases (655 pairs of LE and REC images). This test set was selected to include only Baheya’s dataset, firstly because this dataset has richer clinical CEM information and secondly because all annotations have been performed by CEM expert radiologists. The test set was stratified based on the pathology outcome, with 75 malignant cases, 50 benign biopsied cases, and 25 normal cases. It is important to note that the contralateral breast may exhibit a finding. The pathology per breast is therefore 75 malignant breasts, 66 benign biopsied breasts and 159 non-biopsied breasts. A summary of all clinical information for the test set is shown in Table 2. The table shows clinical information at the case level, including breast density, BPE, and radiological BIRADS, as described in the CEM BIRADS lexicon [30]. The Table 2 also includes annotation length. This length represents the size of the entire disease and not each individual cancer nodule.

The remaining dataset was split into a train set of 787 patients and a validation set of 150 patients. They were stratified in the same way as the test set, i.e., based on the pathology outcome (malignant cases, benign biopsied cases, and normal cases).

## 4. Methods

### 4.1. Deep Learning Model

The proposed CEM CAD model is based on a YOLOv5 architecture which is a single-stage object detection deep learning mode [31,32]. The YOLOv5 architecture has three main components:A backbone network: a feature extraction network that takes the input images and extracts high-level features. The input image is progressively down-sampled, while the number of channels increases, resulting in feature maps with reduced spatial dimensions and increased depth. The backbone architecture is a CSPDarknet [33,34] and size depends on the model scale chosen. The different model sizes are refereed to as “small”, “medium”, “large”, summarized by the indexes “s”, “m”, and “l”, when considering, respectively 53, 71, and 89 convolutional layers and 256, 512, and 512 output channels.A neck: a pyramidal architecture used to merge features from different levels of the backbone network. The YOLOv5 model uses a PAN (path aggregation network [35]) neck to combine feature maps of different resolutions obtained from the backbone and generates a fused multi-scale representation. The multi-scale characteristics of this backbone ensure an optimized detection of all the lesion sizes and aspect ratios. The final merged multi-scale feature map size before the head is, for the “s”, “m”, “l” models, respectively, 26 × 26 × 512, 26 × 26 × 1024, and 40 × 40 × 1024.A head, which is responsible for performing object detection on the features extracted by the backbone network. The YOLOv5 model uses a YOLO head that predicts bounding boxes, and class probabilities for each cell. For our application: the final layer is composed on 3 classes related to the clinical outcome: malignant, biopsied benign, and non-biopsied findings. The ground-truth labels for those classes are obtained from the radiologist annotation, supported by the radiology/pathology reports. While trained on those 3 classes, in the proposed study, only the malignant class results will be quantified as the CAD goal is to detect cancers.

After a series of experiments, the distance intersection over union (DIoU) was selected as a loss function [36] for this study. DIoU encourages more accurate localization of objects. It combines the intersection over union (IoU) and the distance to the center of the ground-truth boxes. The IoU determines the ratio of the overlap area between the detected and ground truth bounding boxes to their union area. The IoU loss alone is difficult to optimize during the first epoch when the model is not trained enough as the intersections between the ground truth and detected box may be null with zero gradient. On the opposite side, the distance term provides a wider convex potential. Gradient-based optimization algorithms are more likely to converge to the global minimum, resulting in improved training and better localization accuracy. The $L_{\mathrm{DIoU}}$ loss function is defined as
(1)$$L_{\mathrm{DIoU}}=1-\mathrm{IoU}+\frac{\rho^{2}\left(x,x^{gt}\right)}{c^{2}},$$
where ρ denotes the Euclidean distance between the coordinates of the predicted and ground-truth boxes’ centers: respectively, x and $x^{gt}$. *c* is the diagonal length of a rectangular box circumscribed to the predicted and ground-truth detections. Our experiments showed that the DIoU loss function leads to better convergence than the traditional IoU loss function. The DIoU loss was evaluated towards other similar losses (e.g., IoU, complete IoU, generalized IoU) and provided better results on all metrics and convergence speed. In addition to this detection loss, the classification loss is a cross-entropy loss, which evaluates the correctness of predicted class probabilities.

To mitigate the challenge of limited datasets, two approaches were employed: data augmentation and transfer learning. In particular, data augmentation strategies suitable for breast images were applied, which included horizontal image flips, global intensity transforms (saturation, brightness changes), and dedicated breast geometrical realistic transforms [37]. The realistic breast geometric transforms represent a set of displacement fields mimicking the motion between multiple acquisitions of the same breast. Those transformations create homotopic breasts with enhanced variability while preserving the breast texture and the pathology outcome. The model weights were pre-trained on Image-Net and subsequently fine-tuned on the lesion detection task. The pre-trained generic model can be found in the Github repository: https://github.com/ultralytics/yolov5 accessed on 26 June 2023.

Finally, two inference techniques were leveraged to improve the accuracy and robustness of the predictions: test time augmentation (TTA) and model ensembling is shown in Table 3. TTA involves duplicating the input image and applying invarience/equivarience such as scales and flips to each copy before combining all detections obtained from the duplicates. Model ensembling consists in training multiple models with different architectures on the same dataset and averaging the outputs. Five models with different hyperparameter choices were trained on the same datasets and averaged to give our final CEM CAD model.

The model has been trained to classify, from a single view input image, three distinct categories: cancer, benign biopsied lesions, and non-biopsied findings. However, the scope of this paper is limited to evaluating the model’s performance in detecting cancer exclusively. Therefore, only the score associated to the cancer category is then considered for this study. Following the inference, a class-agnostic non-maximum suppression (NMS) technique is implemented to decrease the amount of overlapping detections.

Stochastic gradient descent (SGD) was used for optimization with a batch size set to 12 images to fit the 24Gb—Quadro RTX-6000 GPU memory. A complete training took approximately 500 iterations to reach convergence in 8–12 h training time.

RAW LE and HE CEM images were recombined using the latest available recombination algorithm: SenoBright™ HD with NIRA giving an optimal CEM quality recombination images [38].

In this study, we evaluated the CEM CAD performance considering various inputs scenario:REC image only;LE and REC images, as performed by radiologists during the standard clinical practice.

### 4.2. Evaluation Metrics

The evaluation of the CEM CAD can be performed at different levels [39]: (1) cancer detection metrics and (2) breast classification metrics.

#### 4.2.1. Detection Metrics (Lesion Level)

To validate the detection of a ROI, an acceptance criterion based on the Intersection over Union (IoU) between the detected cancer class and the ground truth cancers has been used. This criterion allowed us to assess the accuracy of the detection by measuring the spatial overlap between the predicted bounding boxes and the actual ground truth bounding boxes. The threshold is set to IoU${}_{\mathrm{threshold}}\;=\;0.1$ and allows us to compensate slight annotation inconsistencies [40]. All detections below this threshold are not considered positive.
FROC and AUFROC—The results for the enhancing cancer detection are evaluated with FROC (free receiver operating characteristic) curves considering the sensitivity for the detection of cancer boxes (Se${}^{l}$, defined per lesion) with respect to the number of false positives per image (FPR).The different results will be quantified using an area under the normalized FROC curve, AUFROC(ζ), up to a certain FPR threshold ζ.
(2)$$\mathrm{AUFROC}\left(\zeta \right)=\frac{1}{\zeta }\int_{0}^{\zeta }{\mathrm{Se}}^{l}\left(\mathrm{FPR}\right)\;d\mathrm{FPR},$$
ranging in [0, 1]. To ensure comparability, the ζ threshold for different computations must be identical. The standard value taken for the results section is ζ=0.3. This metric is used to evaluate the effectiveness of a CAD system in reducing unnecessary biopsies of benign lesions. For this metric, the sensitivity is computed at the lesion level.Quality of the detections—The IoU quantify the spatial accuracy to accept a detection. The threshold value depends on the imaging modality, on the initial annotation variability, and on the final expected accuracy of the output. The detection results at different IoU thresholds will be presented.Detection heatmap—A heatmap was generated to show the probability of detection. This heatmap displays all detections (without NMS) using a Gaussian function with anisotropic characteristic lengths that are equal to the size of the rectangular detection. The amplitudes of these Gaussian functions are the confidence scores. Areas with a high probability of being cancerous are depicted in red values, while areas with a high probability of being benign are depicted in blue.

#### 4.2.2. Breast Classification Metrics (Breast Level)

The suspicion score predicted for each detection enables a broader classification at the patient, breast, or image level. The suspicion score for each breast is determined based on the highest score of the most suspicious detected box across all available breast views. To evaluate this classification, an ROC curve has been built to measure sensitivity Se${}^{b}$ and specificity Sp${}^{b}$ (with the index *b* denoting the breast level). The CEM CAD performance was compared with those from radiologists in CEM diagnostic clinical review papers [41,42]. The evaluated metric will be the standard area under the receiver operating characteristic (AUROC):(3)$$\mathrm{AUROC}=\int_{0}^{1}{\mathrm{Se}}^{b}{(1-\mathrm{Sp}}^{b})\;d{\mathrm{Sp}}^{b}.$$

For this metric, bootstraping was used to compute the 95% confidence interval (CI).

## 5. Results

### 5.1. Evaluation of Different Inputs Channels Impact

The cancer detection and breast classification results for different input images (LE and REC images, and REC only) are presented in Figure 1. The thin lines represent individual models, whereas the ensemble model performance is represented in thick lines.

The ensemble model’s AUFROC (0.3) for cancer detection, presented in Figure 1a, is 0.733 (red curve) and 0.853 (black curve) for REC only and the LE and REC coupling, respectively.

For breast classification, presented in Figure 1b, the CEM CAD’s AUROC is 0.916 [CI-95%: 0.886–0.945] (red curve) and 0.964 [CI-95%: 0.946–0.979] (black curve) for REC only and the coupling of LE and REC, respectively.

For both evaluation metrics (at lesion level and breast level), the best performance is obtained when considering both LE and REC images as inputs for the CAD model. The addition of the LE information allows increasing the cancer sensitivity results of around 0.1 over the whole FROC curve. Therefore, we used LE and REC images as inputs for all results presented below.

### 5.2. Enhancing Cancer Detection Results (Lesion Level)

Once the CAD detects a box as a potential cancer, it is essential to determine the likelihood of it being a true cancer. Figure 2 presents the percentage of cases in which the detected box corresponds to a ground truth cancer with respect to the sensitivity at the lesion detection level. Additionally, the figure displays the class of the detected box if it is not a true cancer. The orange curve represents benign biopsied lesions, while the black curve represents unsuspicious detection (non-biopsied texture).

For sensitivity under 0.5, all false positives are benign biopsied lesions. In the range of Se${}^{l}$ between 0.65 and 0.92, the false positives are equally represented by benign biopsied lesions and other findings. When the cancer sensitivity increases after 0.92, the CAD detects mainly unsuspicious (non-biopsied) textures as the dark curve highly increases.

The CEM CAD results for two patients (eight treated views) are presented in Figure 3. Red and green boxes represent ground-truth cancers and benign lesions. Blue boxes are the CEM CAD cancer detection with the CAD suspicion scores ranging in [0–1]. While no CC/MLO consistency is imposed, the same lesion detection is found in the two views (similar distance-to-nipple and lesion sizes). For the patient presented on the bottom, two areas are detected in the RCC view, while those identified by a radiologist are treated as a single, large box.

#### 5.2.1. Detection Heatmap

The heatmap (Figure 4), presented for various patient breasts, shows the areas detected by the algorithm, where red indicates high probability of cancer and blue indicates high probability of benignity. The algorithm identified different areas, including a large contrast uptake in (a) and (b) highlighted in red, axillary nodes in (a), (c), and (e) shown in blue, and a benign contrast uptake in the upper quadrant in (c). The area of suspicion in (d) is very large and matches with the large disease.

#### 5.2.2. Detected IoU

The IoU threshold plays an important role in assessing if a detected area is correctly located or not. Figure 5 shows the FROC curves at different IoU thresholds. With a higher acceptance threshold, the sensitivity at the lesion level decreases. The AUFROC (0.3) for an IoU threshold of [0.01, 0.05, 0.1, 0.15, 0.2, 0.3] are [0.874, 0.871, 0.853, 0.814, 0.779, 0.726], respectively. Two curves are highlighted, with an IoU threshold of 0.05 (red curve) and 0.1 (plain black curve). At an IoU threshold of 0.1, the mean IoU of the detected lesion is high (around 0.44); hence, the lesion surfaces are well detected. This corresponds to, for example, large, clearly malignant contrast uptakes.

With a sensitivity of 0.90, the trained model in this study performed cancer detection at 0.085 false positives per image at an IoU threshold of 0.05 (compensating for annotation inconsistency) or 0.128 false positives per image for an IoU threshold of 0.1.

The IoU threshold does not impact the breast classification and ROC curves as the box acceptance is not included in the breast classification strategy. The output of such inconsistencies introduces many false positives and false negatives, as shown in Figure 6. In this figure, the two small detected boxes in the left cranio-caudal view (depicted by the arrows) have an IoU with the ground truth smaller than 0.03 and are thus considered as false positives with the discussed IoU threshold.

### 5.3. Breast Classification Results (Breast-Level)

Breast classification results are presented in Figure 1b. The CEM CAD’s AUROC for the final ensembling model is 0.964 [CI-95%: 0.946–0.979] (black curve). The AUROCs for the five models are 0.967, 0.968, 0.969, 0.974, 0.966, respectively, for models “5s-(a)”, “5s-(b)”, “5m-(a)”, “5m-(b)”, “5l”.

The disks, corresponding to radiologists’ diagnostic results, are extracted from clinical review papers [41,42]). The size of the disk represents the size of the the clinical dataset. The median specificity for the breast classification in the 24 clinical studies is 0.744. At this fixed specificity, the sensibility of the AI model considering LE and REC for the breast classification is Se${}^{b}$ = 0.988. As a comparison, the sensitivities reported in the reviews are, respectively, for the min/median/max: 0.543/0.952/1.00.

### 5.4. Evaluation of the BPE Impact

The detection and classification results can be split in two sub-categories of background parenchymal enhancement level: low (104 cases–437 images) and high (46 cases–197 images), corresponding, respectively, to the grades minimal/mild and moderate/marked. The results for low (104 cases, 437 images) and high (46 cases, 197 images) BPE grades are presented in Figure 7.

The FROC and ROC curves for low (104 cases, 437 images) and high (46 cases, 197 images) BPE levels are presented in Figure 6. The model’s AUFROC(0.3) for cancer detection is 0.891 (red curve) and 0.712 (blue curve) for low and high BPE, respectively. The AUROC for breast classification is 0.986 (0.976–0.995) (red curve) and 0.919 (0.866–0.962) (blue curve) for low and high BPE, respectively. These results show that the algorithm’s performance is lower for high BPE compared to low BPE.

All detection and classification results are summarized in Table 4. The comparison of the performance with other published CEM-CAD models can be performed. The results are compared with previous CAD development with an equivalent architecture [27]. The new proposed method outperforms all previous obtained metrics. It is difficult to compare with the results from [25] as the authors presented pixel (segmentation) metrics and no metrics at the lesion level. For the breast classification metric, the results are in the range of the model developed by Zheng et al. [28]. In their paper, they obtained an AUROC of 0.947 (0.916—0.978), evaluated on their internal testing set with an Asian population.

## 6. Discussion

### 6.1. Input Data Impact

This study shows that, as radiologists do, leveraging REC and LE image channels improves the performance of the CEM CAD. In addition, this results is in line with the development of region-of-interest CEM classification models [43]. The low-energy images complement the recombined images and and enable better discrimination of cancers.

### 6.2. Lesion Detection and Breast Classification

The results presented in Table 4 show that the different models used for ensembling have similar performance on cancer detection and breast classification, indicating that the choice of architecture has a limited impact on the results. The ensembling does not improve the metrics (AUROC, AUFROC, and the confidence interval) compared to each individual model. This suggests that the learnt content may be similar. The obtained results validate the chosen size of the 150-case test dataset [44] (high AUC: AUC > 0.9, small AUC difference between the tests: <0.05 and five algorithms).

The heatmaps presented in Figure 4 show different areas detected as benign and malignant. With this representation, the model successfully identifies lymph nodes, benign lesions, as well as cancers.

One main difficulty in this work lies in the consistency of the lesion annotation. The IoU threshold for detection acceptance with the ground truth has a significant impact on the metrics, as shown in Figure 5. A low IoU compensates for slight inconsistencies in data annotation as data come from various sites with different annotators. For instance, a multi-focal carcinoma can be identified by a large box or by annotating each individual nodule. While being very close to the main lesion, those detected nodules could be merged and count as a single, larger detection.

The results of the proposed model outperform previously published detection CEM-CAD and could be linked to a larger training dataset. For the breast classification, the proposed CAD model obtained similar performance to Zheng et al. [28] yet considered complex cases such as multiple, bilateral, and non-mass lesions and lesions seen only in one view. Comparing metrics in detail with results from other papers may be challenging due to significant differences in the datasets. These variations are on diverse populations, clinical characteristics, and inclusion/exclusion criteria. Consequently, direct comparisons between studies may not be straightforward or yield precise conclusions. This emphasizes the importance of conducting studies with standardized protocols and larger, more representative cohorts to ensure more robust and reliable comparisons.

### 6.3. BPE Impact

The study showed that the BPE level impacts CEM CAD results. This is in line with previous clinical studies which suggested that increased BPE on CEM may lead to increases in false-positive rates [45]. In diagnostics, distinguishing between BPE and lesion contrast uptake is challenging [46]. Therefore, it is important to collect data that include high BPE cases, which are more difficult to diagnose. Due to the low number of cases in the marked and minimal BIRADS categories, the study was unable to split the data into these four categories. It is therefore crucial to consider BPE when evaluating the performance of CEM CAD. Future research should focus on optimizing the system for high BPE cases.

### 6.4. Limitations and Improvement Opportunities

The developed study has several limitations and improvement opportunities. First, this study was performed with mainly trained and fully evaluated data coming from a unique site (Baheya Foundation). This particularity may generate a dataset/annotation bias in the results when applied to other sites. Therefore, collecting data from more than one site will ensure the model generalizes on various populations. Additionally, merging multiple site data, such as multiple patients, lesions, and clinical scenarios, can further improve the performance of the CEM CAD.

The data utilized in this study were annotated by a single group of experienced radiologists, which could introduce annotation bias. To address this concern and reduce reader variability, it is essential to annotate each case with multiple radiologists. Implementing a consensus/merging strategy to obtain a ground-truth annotation would be a necessary step.

One assumption in this study is that all non-biopsied areas are assumed to be benign/normal (hence negative for malignancy). This is a strong hypothesis as a non-detected cancer by radiologists could have been detected by the AI. To exclude the possibility of a malignant lesion developing, a standard consists in excluding all cases without available (at least) 6-month negative follow-ups. In the current dataset, more than half of the non-biopsied cases have a negative follow-up; however, this is not the case for all instances.

The recent development of full-field digital mammography (FFDM) detection CAD systems could complement this work [47]. FFDM data are available in much larger quantities than CEM data. Those FFDM-CAD have demonstrated high efficiencies and could be applied as a baseline on the LE images of CEM exams. The FFDM-CAD could hence be applied to the LE image and coupled with the proposed CEM CAD model. The results of this study could improved the discrimination of enhancing malignant and benign lesions, especially on micro-calcifications.

Finally, the current algorithm is applied to each image independently without considering the whole exam information. The recent development of multi-view CAD in FFDM has demonstrated significant improvements in results, from a simple average of the outputs [28] to more complex multi-view architectures [48] leveraging the consistency in texture and lesion characteristics between the CC/MLO and L/R views. Exploring this consistency in contrast-enhanced mammography (CEM) will help to reduce false positives.

## 7. Conclusions

In this study, a deep learning model for enhancing cancer detection in contrast-enhanced mammography images are introduced. The model was trained, validated, and tested using an important CEM dataset comprising 1673 patients, totaling 7443 images. These images were collected from different hospitals and acquired through various imaging systems from GEHC.

The proposed detection model is based on the YOLOv5 architecture, trained with data augmentation, and uses inference optimization strategies. First, our analysis demonstrated that the best results where obtained when using both LE and HE images as inputs for the CEM CAD model. Diverse metrics were employed to evaluate the efficiency of the lesion detection and breast classification. The trained model achieved breast classification with an AUROC of 0.964 and was able to identify each individual cancer at a sensitivity of 0.9 and 0.128 false positives per image. These results outperform previous published papers and are comparable with the radiologists’ efficiency reported in the literature. Finally, the BPE level impact investigation highlighted the strong influence of BPE on the CEM CAD results for diagnostic applications.

Different opportunities have been identified to further improve our CEM CAD model performance, such as collecting more high BPE images, increasing model robustness to annotation inconsistency, or implementing a multi-view strategy. Overall, the model developed in this study has the potential to assist radiologists in detecting and characterizing complex contrast uptake and classifying breast suspicion. With promising results, we aim to evaluate shortly the clinical impact of a CEM CAD in a diagnostic setting.

## Figures and Tables

**Figure 1 bioengineering-10-00974-f001:**
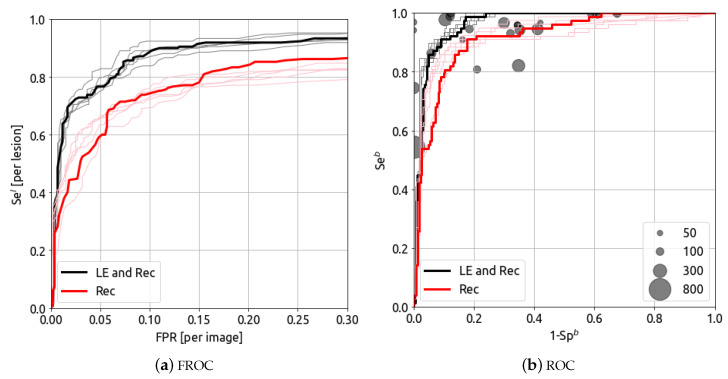
Impact of the image information (**a**) FROC curves with different input channels and (**b**) ROC curves with different input channels. The thin lines represent individual models, whereas the ensemble model performance is represented in thick lines. The disks represent radiologists’ diagnostic results from [41,42]. The size of the disk represents the size of the the clinical dataset.

**Figure 2 bioengineering-10-00974-f002:**
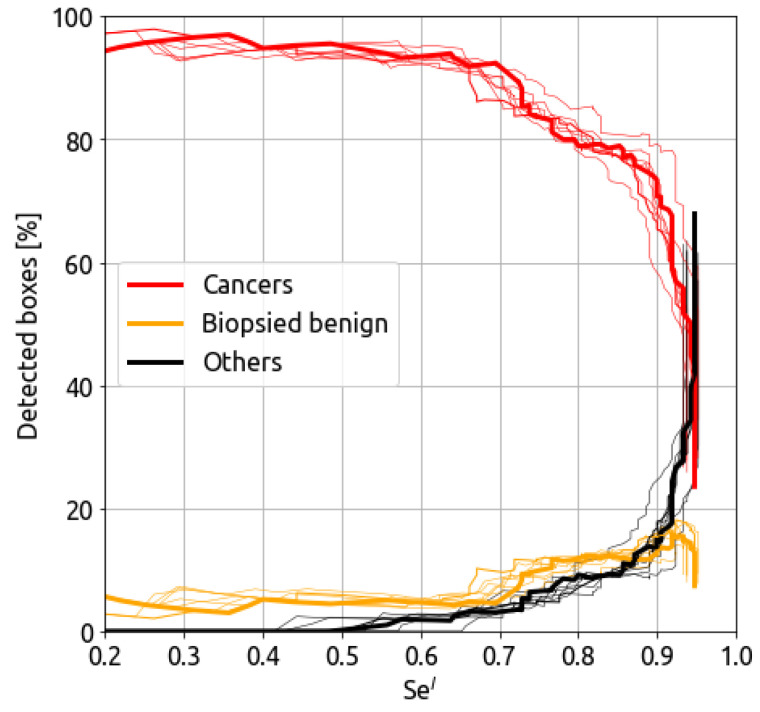
Ground-truth probability for the detected cancer boxes with respect to the cancer detection sensitivity Se${}^{l}$. The thin lines represent individual models, whereas the ensemble model performance is represented in thick lines.

**Figure 3 bioengineering-10-00974-f003:**
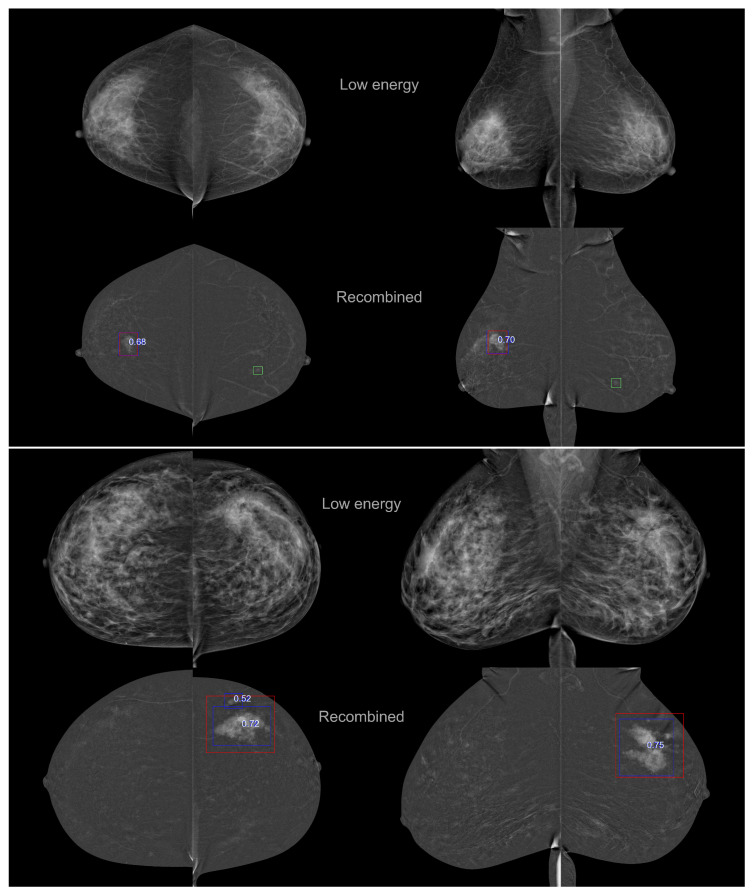
CEM CAD detection results (blue boxes) and detection scores for two patients’ CEM exams. The first lines correspond to the low energy images and the second line to the recombined images. Red and green boxes represent ground-truth cancers and benign lesions, respectively.

**Figure 4 bioengineering-10-00974-f004:**
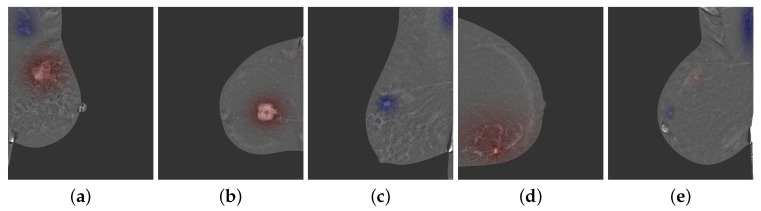
Detection heatmap for different images. Red areas (in (**a**,**b**,**d**)) represents high probability of cancer, while blue areas (in (**a**,**c**,**e**)) represent high probability of benign finding.

**Figure 5 bioengineering-10-00974-f005:**
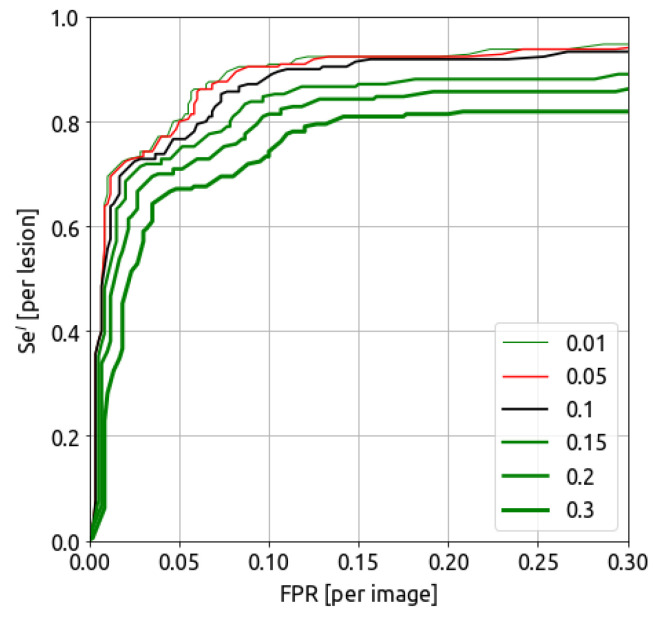
FROC curves considering different IoU thresholds.

**Figure 6 bioengineering-10-00974-f006:**
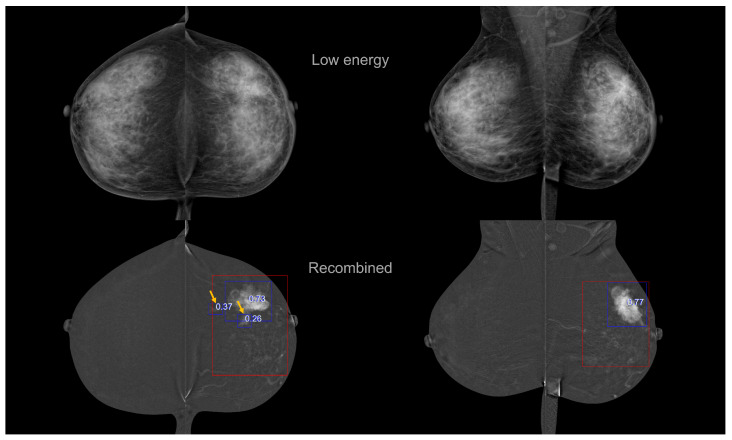
Example of detection inconsistency with the annotation. The red and blue boxes are, respectively, ground-truth cancer annotation and CAD detected areas. In this cases, the small detected enhancing areas in the left cranio-caudal view are considered as two false positives.

**Figure 7 bioengineering-10-00974-f007:**
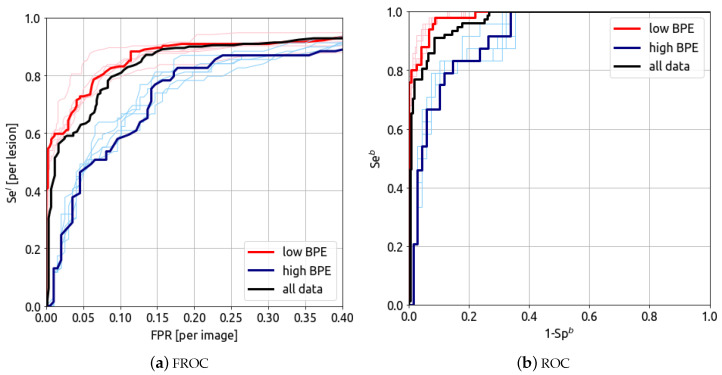
(**a**) FROC curve and (**b**) ROC curve for different categories of BPE. Red: low BPE (minimal and moderate grades). Blue: high BPE (moderate and marked grades).

**Table 1 bioengineering-10-00974-t001:** CEM dataset and pathology results. The presented values are the number of patients. The wording “images” refers to a pair of low-energy and recombined information. * proven by tissue diagnosis.

Institution	Number of Patients	Acquisition System	Pathology Results		
			Non-Biopsied	Benign *	Malignant *
Baheya Foundation For Early Detection And Treatment Of Breast Cancer (Egypt)	1087 (4933 images)	Pristina	136 (13%)	199 (18%)	752 (69%)
Peking University First Hospital & Shanghai First People’s Hospital (China)	244 (976 images)	Essential/DS	35 (14%)	64 (26%)	145 (60%)
University of Washington (US)	187 (790 images)	Essential	153 (81%)	30 (2%)	4 (81%)
Beth Israel Deaconess Medical Center (US)	50 (211 images)	Essential	2 (4%)	24 (48%)	24 (48%)
Hospital del Mar (Spain)	40 (234 images)	Pristina	0 (0%)	15 (38%)	25 (62%)
University of Cambridge (UK)	39 (156 images)	Pristina	0 (0%)	11 (28%)	28 (72%)
CBIS—Carolina Breast Imaging Specialists (US)	26 (143 images)	Pristina	1 (5%)	5 (19%)	20 (77%)

**Table 2 bioengineering-10-00974-t002:** Clinical information on the test set, according to BIRADS lexicon [30].

Age [Year]	Mean ± Std (Min–Max)	49 ± 10 (27–80)
Breast composition (per case)	A	5/150
	B	86/150
	C	53/150
	D	6/150
BPE (per case)	minimal	28/150
	mild	75/150
	moderate	39/150
	marked	8/150
BIRADS (per breast)	1	107/300
	2	48/300
	3	44/300
	4	60/300
	5	41/300
Pathology (per breast)	Non-biopsied	159/300
	Benign-biopsied	66/300
	Malignant	75/300
Annotation length [mm]	mean ± std (min–max)	35 ±25 (6–170)

**Table 3 bioengineering-10-00974-t003:** Trained models for the ensembling.

Model	Batch-Size
YOLOv5s-(a)	12
YOLOv5s-(b)	12
YOLOv5m-(a)	8
YOLOv5m-(b)	8
YOLOv5l	6

**Table 4 bioengineering-10-00974-t004:** CEM CAD results.

Model	AUROC	AUROC—95% CI	Sp${}^{b}$@Se${}^{b}$ = 0.90	AUFROC (0.3)	FPR@Se${}^{l}$ = 0.90
YOLOv5s-(a)	0.967	0.951–0.983	0.919	0.839	0.148
YOLOv5s-(b)	0.968	0.953–0.983	0.927	0.827	0.230
YOLOv5m-(a)	0.969	0.953–0.984	0.925	0.848	0.148
YOLOv5m-(b)	0.974	0.949–0.982	0.949	0.877	0.080
YOLOv5l	0.966	0.951–0.982	0.929	0.855	0.110
proposed CEM CAD	0.964	0.946–0.979	0.934	0.853	0.128
midrule low-BPE	0.986	0.976–0.995	0.966	0.891	0.080
high-BPE	0.919	0.866–0.962	0.795	0.712	0.416
REC only	0.916	0.866–0.945	0.861	0.733	0.837
Jailin et al. [27]	0.930	-	0.790	0.632	0.410
Zheng et al. [28]	0.947	0.916–0.978	-	-	-

## Data Availability

Not applicable.

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
