# Peer review of "AI-Based Cancer Detection Model for Contrast-Enhanced Mammography"

_bioengineering, 2023, doi:10.3390/bioengineering10080974_

Round 1

Reviewer 1 Report

1.     Abstract suffers from critic issues. In the abstract, the main contributions are usually put into the context of the relevant field, combining objectives, short methodology, and main findings.

2.     The main contributions are not clear in the introduction, give more details about your contributions.

3.     Add a section for related work, and present previous studies from 2020-2023 with their limitations.

4.     The manuscript briefly mentions the backbone network, neck, and head components but does not delve into the specific details of how they function in the YOLOv5 architecture. Providing more information about the purpose and operation of each component would enhance the understanding of the model.

5.     The manuscript mentions that the final layer of the YOLOv5 head is composed of three classes for the specific application of the CEM CAD model. However, it does not explain what these classes represent or how the model handles them. Adding more details about the specific classification task and how it relates to the CEM CAD model would be helpful.

6.     To show the steps of YOLOv5 it is required to design a Figure.

7.     It is highly required to present the architecture and parameters of YOLOv5 in a table. 

8.     Consider providing a brief explanation of why DIoU was selected and how it contributes to improving the model's performance.

9.     The manuscript briefly mentions data augmentation and transfer learning as two approaches to mitigate the challenge of limited datasets. It would be beneficial to provide more specific details about the augmentation techniques applied, such as the range and type of flips, intensity transforms, and breast geometrical realistic transforms used. Additionally, clarify the source of the pre-trained model weights from ImageNet and explain how they were fine-tuned for the lesion detection task.

10.  Consider providing specific examples of commonly used cancer detection metrics, such as sensitivity, specificity, accuracy, or area under the receiver operating characteristic curve (AUC-ROC), and explain how they are relevant to evaluating the CEM CAD model's performance in detecting cancer.

Author Response

We thank the reviewer for their comments/remarks. The have answered all points in a separate PDF document.

Reviewer 2 Report

This paper presents a deep learning model designed to detect and classify breast lesions using contrast-enhanced mammography (CEM) images. The authors optimized and trained the model on a large dataset of 1673 patients with biopsy-proven lesions. The evaluation of the model's performance was conducted using various metrics, including the Free Receiver Operating Characteristic (FROC) and Receiver Operating Characteristic (ROC) curves. The results showed high accuracy in breast classification and lesion detection, outperforming previously published papers and demonstrating comparable performance to radiologists' diagnostic capability.

Strengths:

- Large Dataset: The use of a substantial dataset consisting of 7443 images from 1673 patients enhances the credibility and generalizability of the model's performance.

- Performance Metrics: The evaluation of the model's performance using established metrics like AUROC and sensitivity provides a quantitative assessment of its capabilities.

- Comparative Analysis: The paper compares the model's performance with previously published works, highlighting its superiority and potential clinical relevance.

- Impact of BPE: The study investigates the influence of breast parenchymal enhancement (BPE) on the model's performance, providing valuable insights for diagnostic applications.

Weaknesses:

- Limited novelty: The adopted techniques have generally appeared in previous studies.

- Limited Discussion of Related Works: The paper lacks a comprehensive discussion of recent related works on AI-based detection for enhanced medical imaging ("Adaptive Squeeze-and-Shrink Image Denoising for Improving Deep Detection of Cerebral Microbleeds" in MICCAI'21). Exploring and referencing relevant studies would strengthen the context and contribute to the existing literature.

- Future Improvement Opportunities: While the paper briefly mentions opportunities for further improvement, such as collecting more high BPE images and increasing model robustness, these suggestions are not explored in detail. Elaborating on these potential enhancements would enhance the paper's practical implications.

The quality of English language in the paper is good. 

Author Response

(The authors gave the same response as above.)

Reviewer 3 Report

In this manuscript, a deep learning based cancer detection model is optimized and trained on a large contrast-enhanced mammography (CEM) dataset of 1673 patients (7443 images) with biopsy-proven lesions from various hospitals and acquisition systems. Although the deep learning model, YOLOv5, is not new, application of the model into CEM does have some novelty. In addition, the large dataset is appealing which can support the deep learning model. The manuscript is well written and organized.

1.   Keywords. “Breast cancer enhancing cancer detection” can be improved. For instance, it can be divided into two: “breast cancer” “cancer detection”.

2.   The limitations and future work should be described in the Discussion section.

3.   The titles of the subsections in the Discussion section should be indicated with 5.1, 5.2 … For instance, “Input data impact.” should be changed to “5.1. Input data impact”.

4.   Please change “contrast enhanced” to “contrast-enhanced”.

5.   Institutional Review Board Statement. The approvement number can be provided.

Author Response

(The authors gave the same response as above.)

Round 2

Reviewer 1 Report

1. Related work is essential, thus, it is required to add a Section for related work by presenting some most recently published related work to show the research gap.

2. More numerical results should be provided then compare with some of the previous studies.

3. Highlight the limitations of the proposed method.

Author Response

  1. Related work is essential, thus, it is required to add a Section for related work by presenting some most recently published related work to show the research gap.
  • To the best of our knowledge, we have referred to all existing approaches in the field of CEM-CAD for lesion detection. As a recent expanding modality, very few studies tackle lesion detection in CEM. Although it is not the manuscript format requested by the journal https://www.mdpi.com/journal/bioengineering/instructions, we have followed the reviewer's request and have added a specific section: “Related works in CEM-AI” with 14 relevant recent publications. In addition, we included details of the cited references. We hope this presentation will be clearer and help the reader understand the current research environment.
  1. More numerical results should be provided then compare with some of the previous studies.
  • Multiple numerical results are presented in the paper: ROC, FROC, AUROC, ARFROC, Sensitivity at the lesion level, Sensitivity at the breast level, Specificity (FPR) at the lesion level, Specificity at the breast level, and visualization of the detected areas. Results are presented with uncertainty quantification. We believe that those numerical results allow having a fair evaluation of our model and comparison with other papers. As stated above, few published works are performed on CEM-CAD. The related papers' results were added to the summary table (in the available categories) and discussed.
  1. Highlight the limitations of the proposed method.
  • Following the reviewer’s comment, we have added a specific section for the limitation of the proposed method. We have also put a highlight on the limitation concerning the annotation quality.